# Effects of Supraglottic Airway Devices on Hemodynamic Response and Optic Nerve Sheath Diameter: Proseal LMA, LMA Supreme, and I-gel LMA

**DOI:** 10.3390/medicina59040753

**Published:** 2023-04-12

**Authors:** Rahşan Dilek Okyay, Gamze Küçükosman, Bengü Gülhan Köksal, Özcan Pişkin, Hilal Ayoğlu

**Affiliations:** Anesthesiology and Reanimation Department, Faculty of Medicine, Zonguldak Bülent Ecevit University, Zonguldak 67600, Turkey; gamzebeu@gmail.com (G.K.);

**Keywords:** hemodynamic response, optic nerve sheath, supraglottic airway devices, ultrasound

## Abstract

*Background and Objectives*: Supraglottic airway devices (SADs) are known to be useful in eliminating the drawbacks of laryngoscopy and tracheal intubation, especially ocular pressure and stress responses. The ultrasonographic measurement of optic nerve sheath diameter (ONSD) reflects increases in intracranial pressure (ICP). In our study, we aimed to compare the effects of SADs on hemodynamic response and ONSD. *Materials and Methods*: Our prospective study included 90 ASA I–II patients over the age of 18 who did not have a history of difficult intubation or ophthalmic pathology. The patients were randomly divided into three groups based on the laryngeal mask airway (LMA) devices used: ProSeal LMA (pLMA, *n* = 30), LMA Supreme (sLMA, *n* = 30), and I-gel (*n* = 30). The bilateral ONSD measurements and hemodynamic data of the patients who underwent standard anesthesia induction and monitoring were recorded before induction (T0) and 1 min (T1), 5 min (T5), and 10 min (T10) after SAD placement. *Results*: At all measurement times, the hemodynamic responses and ONSD values of the groups were similar. In all three groups, intergroup hemodynamic changes at T0 and T1 were similar and higher than those at other times of measurement (*p* < 0.001). The ONSD values of all groups increased at T1, and they tended to return to baseline values afterward (*p* < 0.001). *Conclusions*: We concluded that all three SADs could be used safely because they preserved both hemodynamic stability and ONSD changes in their placement processes, and they did not cause elevations in ONSD to an extent that would lead to increased ICP.

## 1. Introduction

The stimulation of the supraglottic region during laryngoscopy and intubation is known to cause a short-lasting (approximately 5 min) elevation in heart rate (HR), blood pressure, intraocular pressure (IOP), and intracranial pressure (ICP) by leading to an increase in plasma catecholamine concentrations through the activation of the sympathoadrenal system [1,2]. Supraglottic airway devices (SADs) have recently become increasingly prevalent in the provision of safe and efficient airways in appropriate conditions. Many studies including adult and pediatric patients have revealed that in comparison to endotracheal intubation, SADs are more beneficial, as they do not require muscle relaxation, are easily applied without laryngoscopy, are less traumatic, and provide more stable hemodynamic and IOP values [3,4,5]. 

The ProSeal laryngeal mask airway (pLMA) (PLMA; Teleflex Medical, Westmeath, Ireland), which was developed by modifying the classic LMA (cLMA) in the 2000s, is a reusable SAD. It includes a hard bite block on the occlusal level and a pilot entry to allow placement using a finger or an “introducer”, which is a metal rod that provides ease of placement. Its placement may require a bit more experience than the cLMA does [5]. The Supreme laryngeal mask airway (sLMA; Ambu**^®^** AuraGain^TM^-Ambu, Ballerup, Denmark), which was developed in 2007, is a modified single-use form of the pLMA. Its preformed bent shaft has a double lumen, including a central lumen for access to the digestive tract sheathed by a flat, oval-shaped airway lumen for reaching the respiratory tract [6]. The I-gel (Intersurgical Ltd., Wokingham, UK) is a relatively new device that is made out of a soft, gel-like, and translucent thermoplastic elastomer designed to not apply pressure on the anatomical structures of the larynx and pharynx with a cuffless, single-use, and narrow-bore gastric drain tube [7]. Studies have reported similar placement times and placement success rates on the first try for all three SDAs [3,4,5,6,7]. 

The optic nerve is part of the central nervous system and is surrounded by cerebrospinal fluid (CSF). With increased ICP, the subarachnoid space, and especially the retrobulbar segment, is affected by the same pressure, which leads to an increase in sheath diameter. This diameter change allows for the indirect measurement of ICP on the optic nerve sheath by transocular ultrasonography (USG) [8,9]. In a study including 31 healthy volunteers and 31 patients with traumatic brain injury, Geeraerts et al. reported the cutoff value for optic nerve sheath diameter (ONSD) measured by transocular USG as approximately 5 mm [10].

Increases in ICP occur due to various causes in anesthesia and surgical procedures [11,12,13,14,15,16,17,18,19,20,21]. Although invasive methods are defined as the gold standard for ICP measurement, it is not possible to use these methods in a standard anesthetic or surgical intervention [22,23,24]. In recent years, interest in noninvasive methods that are also easily applied in the operating room has increased. The most prominent of these methods is ONSD measurement by transocular USG, and several studies have been conducted on its application [11,12,13,14,15,16,17,18,19,20,21]. The purpose of our study was to compare the effects of three different SADs on hemodynamic response and ONSD.

## 2. Materials and Methods

### 2.1. Compliance with Ethical Standards

After receiving permission from Zonguldak Bülent Ecevit University Clinical Research Ethics Committee (protocol No: 2019-96-12/06, ClinicalTrials.gov Identifier: NCT05499754), this prospective randomized study was conducted in Zonguldak Bülent Ecevit University Hospital, Turkey, between July 2019 and 2020. Informed written consent was obtained from all patients. The flow diagram according to CONSORT guidelines is provided as Figure 1 [25]. 

### 2.2. Patient Population

Ninety patients (aged 18–65 years) in the American Society of Anesthesiologists (ASA) class I–II who were scheduled for elective non-ophthalmic procedures that would last 1–2 h under general anesthesia in the supine position were included in this study. The sample of the study excluded patients with Mallampati and those in ASA class ≥III, and those with a history or suspicion of a difficult airway, more than three SAD placements, past intracranial/ocular surgery, diabetic neuropathy, cerebral edema or elevated ICP, glaucoma, potentially full stomachs, uncontrolled hypertension, obstetric conditions, and a lack of agreement to participate in the study. The demographic data, Mallampati scores, and ASA classes of all patients were recorded.

### 2.3. Application of General Anesthesia and Monitoring

All patients were instructed to fast for at least 8 h before surgery. No premedication was given. When the patient was in the anesthesia room, their head was placed on a soft, 7 cm high pillow before the induction of anesthesia, with their neck flexed and head extended. The patients were connected to electrocardiography, noninvasive blood pressure measurement, peripheral pulse oximeter, end-tidal carbon dioxide (EtCO_2_) measurement, and bispectral index (BIS; Aspect Medical Systems, Newton, MA, USA) measurement devices. After the patients were preoxygenated with 100% O_2_ for 3 min, 1.5 mcg/kg fentanyl citrate (Talinat^®^, Vem ilaç, Ankara, Turkey), 1.5 mg/kg lidocaine (Jetmonal %2^®^, Osel ilaç, Istanbul, Turkey), and 2–2.5 mg/kg propofol (Propofol 1% Fresenius^®^, Fresenius Kabi, Bad Hamburg, Germany) were intravenously (iv) administered. No neuromuscular agent was administered. Until the appropriate depth of anesthesia (BIS, 40–60) for device insertion was achieved, an additional propofol bolus (0.5 mg/kg) was planned.

Each patient was randomly assigned to one of three groups, Group P (pLMA, *n* = 30), Group S (sLMA, *n* = 30), or Group I (I-gel, *n* = 30), using the sealed envelope technique. The size of the SAD was selected based on the patient’s weight, as recommended by the manufacturer. For the sLMA and pLMA groups, the patients weighing 30–50 kg, 50–70 kg, and >70 kg underwent their procedures with sizes 3, 4, and 5, respectively. For the I-gel group, the recommendations included the use of sizes 3, 4, and 5 for weights of 30–60 kg, 50–90 kg, and >90 kg, respectively. When the BIS values of the patients who were being ventilated through a facemask reached the range of 40–60, the suitable SAD was placed by the same anesthesiologist doctor. The pLMA and sLMA were completely deflated before insertion. The patient’s head was placed in a sniffing position. After the occurrence of post-induction apnea, the chosen SADs were lubricated and inserted based on the instructions of their manufacturers. The sLMA and the I-gel were inserted using the rotational technique, whereas the pLMA was inserted with the index finger. The cuffs of the pLMA and sLMA were inflated to the recommended volumes, and checks were undertaken for the presence of an air leak sound coming from inside the mouth. After SAD insertion, the observation of fluctuations in capnography, bilateral chest movements, and stridor absence were determined as criteria for successful insertion. In the case of insufficient ventilation after two attempts, insertion was accepted as unsuccessful, the practitioner was free to decide on the use of an alternative airway device, and these patients were excluded from the study. The duration of SAD insertion (the time between the moment of picking the device to be placed and the moment of observing EtCO_2_ waves) was recorded. For the maintenance of anesthesia, the respiratory volume was adjusted to 6–8 mL/kg, the respiratory rate was set at 12/min, and controlled mechanical ventilation was provided with a 50/50% O_2_/Air mixture, as well as 1–2% sevoflurane (Sevorane^®^, Abbvie, Queenborough, UK) and 0.05–0.2 mcg/kg/min remifentanil hydrochloride (Ultiva^®^ GlaxoSmithKline Manufacturing S.p.A, Parma, Italy) infusion.

### 2.4. Data Management

The ONSD values were measured by an experienced anesthesiologist who was not informed about the study using an Esaote MyLab 30 GOLD^TM^ USG device when the patient was in the supine position. For the measurements, a sterile, water-soluble gel was rubbed onto the upper eyelid, which was in the closed position. The 18–6 MHz linear USG probe was carefully placed onto the upper eyelid covered in the gel. Without applying excess pressure, the part of the optic nerve entering the ocular globe was imaged on the screen. To measure ONSD, after finding the most suitable contrast between the retrobulbar echogenic adipose tissue and the vertical hypoechogenic band, the image was frozen. Using an electronic caliper and placing the cursor on the outer contours of the optic nerve 3 mm posterior of the papilla, transverse measurements were taken. The gel on the eyelid was cleaned off after the measurements were completed. The average of the three measurements was determined as the ONSD value, and ONSD values of ≥5 mm were considered indicative of elevated ICP. 

The mean arterial pressure (MAP), HR, and ONSD values of the patients were recorded before induction (T0), 1 min after SAD placement (T1), 5 min after SAD placement (T5), and 10 min after SAD placement (T10). The same observer, who did not have any information about the study, recorded all the parameters that were examined in the study.

### 2.5. Statistical Analysis

The planned sample size required to detect 95.9% test power (1 − β), 95% confidence (1 − α) and effect size d = 1.16 was 10 people per group. We included 30 patients in each group to compensate for patient dropouts [16]. Data was analyzed by using the Statistical Package for the Social Sciences version 23.0 (IBM SPSS Inc., Chicago, IL, USA) program. Normal distribution assumptions were tested using the Shapiro–Wilk test. One-way analysis of variance (ANOVA) was used to compare the normally distributed independent data. The Kruskal–Wallis H and Friedman tests were used to compare the non-normally distributed dependent data. The Friedman test was used for the intragroup comparisons of the non-normally distributed data. Multiple comparisons were made with Bonferroni correction. Chi-squared tests were used to compare the categorical variables. The normally distributed data are presented with mean ± standard deviation values, whereas the non-normally distributed data are presented with median (minimum-maximum) values. The categorical data are given with frequency (percentage) values. *p* < 0.05 was taken as the level of statistical significance.

## 3. Results 

Clinical and Demographic Characteristics

Ninety patients were included in the study, and ninety completed the study (Figure 1). In all patients, the expected BIS levels were achieved without the need for additional propofol administration, and no patient was excluded due to insufficient anesthesia depth. The SADs were placed in all patients on the first attempt without a problem. There was no statistically significant difference among the three groups in terms of their demographic characteristics, ASA-Mallampati scores, or SAD insertion times (*p* > 0.05) (Table 1).

Whereas there was no significant difference among the HR values of the groups (*p* > 0.05), in the intragroup comparisons, all groups showed significant differences in their HR values measured at different times (*p* < 0.001) (Table 2).

Likewise, whereas there was no significant difference among the MAP values of the groups (*p* > 0.05), in the intragroup comparisons, all groups showed significant differences in their MAP value measurements (*p* < 0.001) (Table 3).

An increase in ONSD was seen in all groups at measurement time T1 right after the insertion of the SADs, whereas there was no significant difference among the groups (*p* > 0.05) (Table 4 and Table 5). The changes in the bilateral ONSD values at each measurement time are given in Table 4 and Table 5.

## 4. Discussion 

In our study, when they were inserted into adult patients with normal airways, the effects of the pLMA, sLMA, and I-gel on hemodynamic responses and ONSD values were similar. In all three groups, the HR and MAP values measured 1 min after SAD insertion were similar to each other, and in the intragroup comparisons, these values were higher than those measured at the other measurement times. In the intragroup comparisons, it was also observed that ONSD values had a tendency to increase 1 min after SAD insertion in all three groups and return to baseline values afterward.

In recent years, noninvasive ICP monitoring methods have become more popular. This is because they have fewer complications compared with invasive methods, are cost-effective, and reduce the need for certain factors, such as expertise in neurosurgery, that require some invasive measurement methods. Ideally, a noninvasive ICP assessment method must be accurate, reliable, pathology-independent, and capable of working in a heterogeneous patient population; use instantly available equipment; and be robust in relation to methodical variations, such as the experience of the operator [26].

The ONSD measurement is subject to error and variation because it is observer dependent. Nevertheless, as the measurement methodology is well-standardized, and the results are repeatable, the variability in ONSD measurement can be minimized [26,27]. Although there is a relationship between ICP and ONSD, the lower threshold value of ONSD in identifying intracranial hypertension has not yet been clearly determined. What has been reported, however, is that the optimum ONSD cutoff value could be anywhere in the range of 4.8–5.9 mm, and strong evidence is needed to identify ONSD values that are required to start treating elevated ICP [9,10,28,29]. In our study, to minimize the differences between measurements, the average values of measurements taken by the same individual were analyzed. We believed that comparing the changes in ONSD values over time would provide more accurate values than using a certain cutoff value of ONSD, and the data were evaluated accordingly.

In recent years, ONSD measurements have also been included in anesthesiology practices [11,12,13,14,15,16,17,18,19,20,21]. Propofol is a drug that is prevalently used in anesthesia, which leads to the constriction of cerebral blood vessels in a dose-dependent manner and causes a reduction in blood flow in the brain, therefore lowering ICP indirectly [12,18]. Although studies comparing the effects of propofol and sevoflurane anesthesia on ONSD have reported different results in the early stages of surgery, the same studies have also shown ONSD changes originating from the Trendelenburg position and pneumoperitoneum in laparoscopic surgeries [16,19,20,21]. It has also been shown that anesthesia depth changes ICP and ONSD values by affecting cerebral blood flow [19]. In a study that examined the effects of three different laryngoscopes on hemodynamic response to laryngoscopy, intubation, and ONSD values, Küçükosman et al. found that none of the three laryngoscopes led to an increase in ONSD to an extent that could indicate elevated ICP [17]. In another study, ONSD changes in robotic laparoscopic surgery were examined, and it was observed that variations in ONSD changed based on age, and that these variations were less pronounced among patients under the age of 63, thus showing that younger patients exhibit better autoregulation for ICP [21]. In our study, by providing the appropriate anesthesia depth along with BIS monitoring, the unwanted effects associated with insufficient anesthesia during SAD insertion were prevented. Additionally, the similar but relatively lower mean age of our patients (42 ± 11 years) compared with those in the studies mentioned above also allowed for degrees of ONSD change that would not result in elevated ICP values. All three SADs examined in our study led to increased ONSD values at the first minute after insertion. However, we believe that these increases are compatible with the sympathetic response associated with LMA insertion.

Supraglottic airway devices are easy to use and atraumatic, and they create minimal somatic and autonomic responses. HR and MAP values increase during insertion and removal, but these differences are lower compared with those in tracheal intubation and extubation [30,31]. In a study, it was shown that cardiovascular responses to laryngoscopy and intubation were not accompanied by ST segment changes [32]. Singh et al. compared hemodynamic responses in 60 anesthetized, paralyzed ASA-I and II patients in whom two different airway maintenance devices were used (i.e., pLMA and I-gel). Changes in MAP values were significantly higher in the pLMA group than in the I-gel patients. The authors concluded that the I-gel was a better choice for a maintenance device than the pLMA due to its ease of insertion and preservation of hemodynamic stability [30]. In a study comparing IOP, HR, and blood pressure values before and after the use of the I-gel, LMA, and endotracheal tubes, Allahyari et al. reported that using an LMA or a tracheal tube led to a more significant increase in IOP values and hemodynamic parameters compared with the I-gel. In this study, it was seen that the hemodynamic changes in the I-gel group were the lowest, and this result may have been caused by its inflatable cuff structure [33]. In our view, the inclusion of patients with Mallampati and ASA scores of <3 and the drug doses that were applied along with BIS monitoring were significant factors in the preservation of hemodynamic stability. Similar hemodynamic responses following the insertion procedures of all three SADs show that our results are compatible with the existing literature.

For the immobilization of patients, their hemodynamic stability, and the early diagnosis and treatment of neurological deficits in endovascular treatments (EVT), practitioners have started to use SADs in anesthesia management. It has been stated that these devices, which are used in EVT to prevent hemodynamic instabilities associated with intubation and extubation, can be a safe alternative in anesthesia management, and there is no complication associated with their usage [34,35,36]. As our study did not involve the use of these devices in patients with neurological problems and those with known or expected ICP elevation, further studies are needed to understand their effects in procedures such as neuroradiological operations.

The literature on SADs is vast, and varying results have been reported on their insertion success rates and times. The main factors that could have affected these results in the methodologies of previous studies have been reported as the experience levels of practitioners and differences in the use of muscle relaxants [3,4,5,6,7,30,33,37,38]. Mukadder et al. investigated the effects of the pLMA, sLMA, and I-gel on insertion time and success and found that the insertion time of the I-gel was shorter, whereas there was no significant difference among the devices in terms of insertion success. Specifically, the insertion success rates of the pLMA, sLMA, and I-gel on the first attempt by anesthesiology residents with the same level of experience were approximately 90% [38]. In our study, no second attempt was needed to insert the SADs into any of our patients, and the insertion times were similar to each other. Nonetheless, it is worth noting that the time it took us to insert the SADs in our study was short. Insertion time can be influenced by the technique used for the device and the definition of insertion time as a parameter. We believe we obtained more objective values because we standardized the management of anesthesia depth, and the interventions were made by the same anesthesiologist.

## 5. Limitations

There were some limitations to this study. One of the most important was that it was not technically possible for the person who recorded the data to be unaware of the airway device being used. Furthermore, as patients with normal airways were included in this study, the results may not be applicable to patients with difficult airways or extended LMA insertion durations. Third, this study was conducted with patients who had low ASA scores, and studies involving those with underlying cardiovascular diseases may provide different results. Another limitation is that we did not measure the ONSD optic nerve sheath diameter just before and after SAD insertion. Finally, the effects of SADs on ICP among patients with intracranial pathologies were not examined. Different results in terms of ONSD may be observed in patients who have a history of elevated ICP or cerebral ischemia.

## 6. Conclusions

We concluded that all three SADs could be used safely because they preserved both hemodynamic stability and ONSD changes in their placement processes, and they did not cause elevations in ONSD to an extent that would lead to increased ICP.

## Figures and Tables

**Figure 1 medicina-59-00753-f001:**
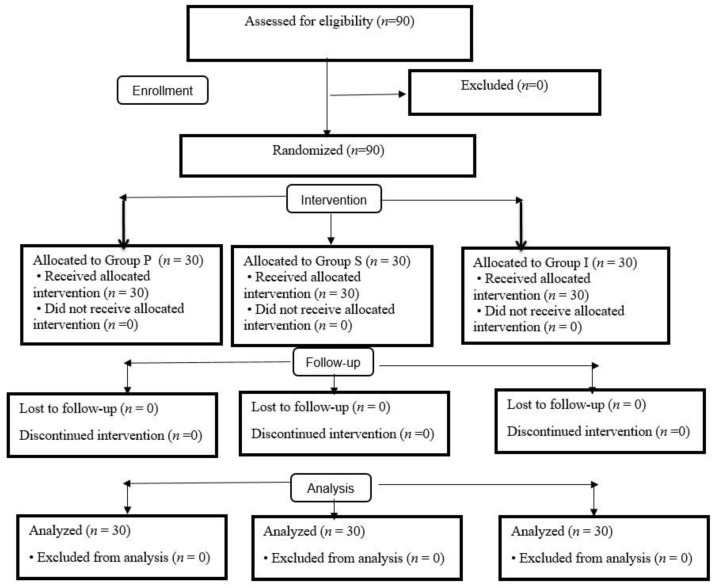
CONSORT flow diagram of the study.

**Table 1 medicina-59-00753-t001:** The groups’ demographic characteristics, ASA-Mallampati scores, and SAD insertion times.

	Group P (*n* = 30)	Group S (*n* = 30)	Group I (*n* = 30)	*p*
**Female/** **Male (*n*)**	24/6	19/11	20/10	0.329
**Age (years)**	42.7 ± 10	43.6 ± 11.5	39.6 ± 12.9	0.386
**Length (cm)**	164.2 ± 8.5	167.5 ± 9.3	168.2 ± 9.3	0.184
**Weight (kg)**	66.6 ± 7.5	66.6 ± 6.2	69.2 ± 7.4	0.248
**ASA (I/II)**	5/25	8/22	5/25	0.535
**Mallampati (I/II)**	10/20	9/21	13/17	0.532
**SAD insertion time (s)**	9.9 ± 3.3	8.2 ± 5.4	7.5 ± 3.9	0.087

Data are presented as *mean ± Standard Deviation or number (n).*
**Group P:** ProSeal laryngeal mask airway, **Group S:** Supreme laryngeal mask airway, **Group I:** I-gel, **ASA:** American Society of Anesthesiologists physical status, **SADs:** supraglottic airway devices.

**Table 2 medicina-59-00753-t002:** Comparison of heart rates within and between groups.

Time (min)	Group P (*n* = 30)	Group S (*n* = 30)	Group I (*n* = 30)	*p* *
**T0**	76 (64–136) ^a^	81.5 (53–116) ^a^	77 (60–132) ^a^	0.783
**T1**	73.5 (55–135) ^a^	74 (52–115) ^a^	77.5 (59–121) ^a^	0.726
**T5**	71.5 (51–105) ^b^	71 (51–108) ^b^	71 (52–110) ^b^	0.976
**T10**	69 (49–100) ^b^	71.5 (50–112) ^b^	67.5 (48–121) ^b^	0.882
***p* ****	**<0.001**	**<0.001**	**<0.001**	

** Kruskal–Wallis test, ** Friedman test.* **Group P:** ProSeal laryngeal mask airway, **Group S:** Supreme laryngeal mask airway, **Group I:** I-gel, **T0:** before induction, **T1:** 1 min after SAD placement, **T5:** 5 min after SAD placement, **T10:** 10 min after SAD placement. * *p*-value: Comparison between groups. ** *p*-value: Compared within the group. a,b: There is no difference between times with the same letter in a group.

**Table 3 medicina-59-00753-t003:** Comparison of mean arterial pressures within and between groups.

Table 30.	Group P (*n* = 30)	Group S (*n* = 30)	Group I (*n* = 30)	*p* *
**T0**	100 (66–136) ^a^	105 (80–137) ^a^	101.5 (73–157) ^a^	0.594
**T1**	92 (59–120) ^a^	85.5 (61–141) ^a^	85 (66–122) ^a^	0.742
**T5**	71 (45–105) ^b^	72.5 (60–125) ^b^	74 (56–122) ^b^	0.641
**T10**	74 (53–121) ^b^	72 (54–98) ^b^	75 (54–110) ^b^	0.360
***p* ****	**<0.001**	**<0.001**	**<0.001**	

** Kruskal–Wallis test, ** Friedman test.* **Group P:** ProSeal laryngeal mask airway, **Group S:** Supreme laryngeal mask airway, **Group I:** I-gel, **T0:** before induction, **T1:** 1 min after SAD placement, **T5:** 5 min after SAD placement, **T10:** 10 min after SAD placement. * *p*-value: Comparison between groups. ** *p*-value: Compared within the group. a,b: There is no difference between times with the same letter in a group.

**Table 4 medicina-59-00753-t004:** Comparison of right-eye ONSD measurements within and between groups.

Time (min)	Group P (*n* = 30)	Group S (*n* = 30)	Group I (*n* = 30)	*p* *
**T0**	3.4 (2.7–4.5)	3.7 (2.4–4.7)	3.5 (2.7–5.2)	0.167
**T1**	4 (3.3–4.9) ^&^	3.9 (2.7–5) ^&^	3.8 (3–5.5) ^&^	0.982
**T5**	3.4 (2.8–4.7)	3.6 (2.7–4.7)	3.7 (2.8–5.3)	0.220
**T10**	3.6 (2.8–4.5)	3.6 (2.5–5.2)	3.5 (3–5.1)	0.845
***p* ****	**<0.001**	**<0.001**	**<0.001**	

** Kruskal Wallis test, ** Friedman test*. **Group P:** ProSeal laryngeal mask airway, **Group S:** Supreme laryngeal mask airway, **Group I:** I-gel, **T0:** before induction, **T1:** 1 min after SAD placement, **T5:** 5 min after SAD placement, **T10:** 10 min after SAD placement. * *p*-value: Comparison between groups. ** *p*-value: Compared within the group. ^&^ *p* < 0.001: Compared with In-Group T1.

**Table 5 medicina-59-00753-t005:** Comparison of left-eye ONSD measurements within and between groups.

Time (min)	Group P (*n* = 30)	Group S (*n* = 30)	Group I (*n* = 30)	*p* *
**T0**	3.5 (2.5–4.3)	3.6 (2.1–4.9)	3.6 (2.8–5.3)	0.303
**T1**	3.9 (3.2–5) ^&^	4 (2.8–5) ^&^	4 (3.1–5.3) ^&^	0.948
**T5**	3.6 (2.8–4.7)	3.5 (2.3–5.7)	3.8 (3–5.1)	0.088
**T10**	3.7 (2.8–4.3)	3.6 (2.8–4.7)	3.7 (3–5.1)	0.707
***p* ****	**<0.001**	**<0.001**	**<0.001**	

** Kruskal Wallis test, ** Friedman test*. **Group P:** ProSeal laryngeal mask airway, **Group S:** Supreme laryngeal mask airway, **Group I:** I-gel, **T0:** before induction, **T1:** 1 min after SAD placement, **T5:** 5 min after SAD placement, **T10:** 10 min after SAD placement. * *p*-value: Comparison between groups. ** *p*-value: Compared within the group. ^&^ *p* < 0.001: Compared with In-Group T1.

## Data Availability

The data used and/or analyzed during the current study are available from the corresponding author.

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
