# Peer review of "Effects of Supraglottic Airway Devices on Hemodynamic Response and Optic Nerve Sheath Diameter: Proseal LMA, LMA Supreme, and I-gel LMA"

_medicina, 2023, doi:10.3390/medicina59040753_

Round 1

Reviewer 1 Report

In this prospective randomized study, the authors compared the effects of three different supraglottic airway devices on hemodynamics and optic nerve sheath diameter. They concluded that heart rate and mean arterial pressure were similar at various timepoints in the intergroup comparisons while at one min after the supraglottic device insertion these values were higher than those measured at other measurement times in the intragroup comparison. Regarding the optic nerve sheath values, these had a tendency to increase one min after the supraglottic device insertion (but not to values indicative of elevated intracranial pressure) and then return to baseline values. Therefore, their effect on optic nerve sheath diameter was mild and temporary.

This work although simple in its conception is of interest and demonstrates that the use of supraglottic airway devices is a useful compromise with less hemodynamic and brain perturbation that laryngoscopy and intubation, therefore their use should become more widespread in anesthesia practice.

My comments are as follows:

1.     Use the format 11-21 when mentioning multiple references instead of mentioning them all. Please check throughout the paper and revise accordingly

2.     Did you exclude patients with potentially full stomachs? You are not mentioning anything like that in the exclusion criteria. Make a comment in the discussion section whether you consider the supraglottic airway devices appropriate for this patient population.

3.     You are mentioning in page 3 a continuous infusion of remifentanil of 0.05 mcg/kg/dk. Do you mean mcg/kg/min? Please revise.

4.     Ii would have made sense to have included one (or two) additional timepoints in your measurements: Immediately preceding the insertion of the supraglottic device (the most important omitted timepoint) and immediately after the insertion. This way, differences between immediately before insertion and following timepoints would have been more obvious and would add merit in the Discussion. Now, you can only talk about differences between 1 min and 5 min post insertion which are not very important in my opinion. Please discuss the lack of these measurements as limitations.

5.     As you correctly point out, temporary increase of optic nerve sheath diameter was not indicative of elevated intracranial pressure as they were well below the value range of 4.8-5.9 mm. This suggests that interventions in the airway are less harmful than we actually believe. I would suggest that you add in the Discussion section a reference suggestive of this which showed that cardiovascular responses to laryngoscopy and intubation are not accompanied by ST segment changes (Theodoraki K, Fassoulaki A. Eur J Anaesthesiol. 2009 Jun;26(6):520-2. doi: 10.1097/EJA.0b013e32831a468d). It would be of interest to see the effect that endotracheal intubation has on optic sheath diameter as compared to the use of supraglottic devices in patients woith normal brain compliance and perhaps this could be the focus of a future study.

Author Response

First of all, thank you for the comments.

We reply all comments as follow.

Reviewer 1

  1. Use the format 11-21 when mentioning multiple references instead of mentioning them all. Please check throughout the paper and revise accordingly

------- We checked throughout the paper and revised.   

  1. Did you exclude patients with potentially full stomachs? You are not mentioning anything like that in the exclusion criteria. Make a comment in the discussion section whether you consider the supraglottic airway devices appropriate for this patient population.

-------   We had already excluded patients with potential full stomach in the study. It has been added to the exclusion criteria as you requested.

           The sample of the study excluded patients with Mallampati and ASA class ≥ III, a history or suspicion of a difficult airway, more than three SAD placements, past intracranial/ocular surgery, diabetic neuropathy, cerebral edema or elevated ICP, glaucoma, patients with potentially full stomachs, uncontrolled hypertension, obstetric conditions, and a lack of agreement to participate in the study.      

  1. You are mentioning in page 3 a continuous infusion of remifentanil of 0.05 mcg/kg/dk. Do you mean mcg/kg/min? Please revise.

-------- Revised as you requested.

  1. I would have made sense to have included one (or two) additional timepoints in your measurements: Immediately preceding the insertion of the supraglottic device (the most important omitted timepoint) and immediately after the insertion. This way, differences between immediately before insertion and following timepoints would have been more obvious and would add merit in the Discussion. Now, you can only talk about differences between 1 min and 5 min post insertion which are not very important in my opinion. Please discuss the lack of these measurements as limitations.

---------- Added to the limitations as you requested.

           ------Another limitation is that we did not measure the ONSD optic nerve sheath diameter just before and after SADs insertion.

  1. As you correctly point out, temporary increase of optic nerve sheath diameter was not indicative of elevated intracranial pressure as they were well below the value range of 4.8-5.9 mm. This suggests that interventions in the airway are less harmful than we actually believe. I would suggest that you add in the Discussion section a reference suggestive of this which showed that cardiovascular responses to laryngoscopy and intubation are not accompanied by ST segment changes (Theodoraki K, Fassoulaki A. Eur J Anaesthesiol. 2009 Jun;26(6):520-2. doi: 10.1097/EJA.0b013e32831a468d). It would be of interest to see the effect that endotracheal intubation has on optic sheath diameter as compared to the use of supraglottic devices in patients woith normal brain compliance and perhaps this could be the focus of a future study.

---------- Added to the discussion section as you requested.

------- In a study, it was shown that cardiovascular responses to laryngoscopy and intubation were not accompanied by ST segment changes (32).

Reviewer 2 Report

Dear authors

there are some spelling errors i.e. "kidrolorur", "sitrat"

The overall conclusion is that the placement of the supraglottic airways is safe but there would be interesting to see what is the effect in case of elevated ICP and ASA III or higher patients, measured with a device that is a gold standard for ICP but  it is invasive, with ventricular or intraparenchymal probes.

The ONSD can provide valuable ICP estimates for the initial non-invasive assessment of patients with suspected raised ICP but there several limitations affecting diagnostic accuracy in detecting raised ICP. Multiple studies reported significant variability in ONSD reference ranges and cut-off values predictive of elevated ICP

Most data come from heterogeneous studies in terms of sample size, inclusion criteria, reference standards and methodological approaches.

The ONSD technique should be standardized.

It is clear that those devices are safe when the anesthesia depth is adequate. I.e. there is a quite intense manipulation with the upper airway during the gastroscopy without serious ICP damage.

Author Response

First of all, thank you for the comments.

We reply all comments as follow.

Reviewer 2

  • there are some spelling errors i.e. "kidrolorur", "sitrat"

------ Necessary corrections have been made.

  • The overall conclusion is that the placement of the supraglottic airways is safe but there would be interesting to see what is the effect in case of elevated ICP and ASA III or higher patients, measured with a device that is a gold standard for ICP but it is invasive, with ventricular or intraparenchymal probes.

------- Thank you for your comment. We mentioned this situation in the limitation section.

  • The ONSD can provide valuable ICP estimates for the initial non-invasive assessment of patients with suspected raised ICP but there several limitations affecting diagnostic accuracy in detecting raised ICP. Multiple studies reported significant variability in ONSD reference ranges and cut-off values predictive of elevated ICP

------ It is more appropriate to prefer more invasive methods in pathological situations, but we excluded patients with a previous history of high intracranial pressure, cerebral edema, and high intracranial pressure before induction.

  • Most data come from heterogeneous studies in terms of sample size, inclusion criteria, reference standards and methodological approaches. The ONSD technique should be standardized.

------  We performed our ONSD measurements with the standard method by experienced anesthetists who had measured at least 100 times before. We avoided making mistakes as much as possible by taking the average value after 3 measurements.

  • It is clear that those devices are safe when the anesthesia depth is adequate. I.e. there is a quite intense manipulation with the upper airway during the gastroscopy without serious ICP damage.

------- Thank you for your comment.
